# Intra- and Inter-Individual Associations of Family-to-Work Conflict, Psychological Distress, and Job Satisfaction: Gender Differences in Dual-Earner Parents during the COVID-19 Pandemic

**DOI:** 10.3390/bs14010056

**Published:** 2024-01-16

**Authors:** Berta Schnettler, Edgardo Miranda-Zapata, Ligia Orellana, Mahia Saracostti, Héctor Poblete, Germán Lobos, Cristian Adasme-Berríos, María Lapo, Katherine Beroiza, Andrés Concha-Salgado, Leonor Riquelme-Segura, José A. Sepúlveda, Karol Reutter

**Affiliations:** 1Facultad de Ciencias Agropecuarias y Medioambiente, Universidad de La Frontera, Temuco 4811230, Chile; 2Centro de Excelencia en Psicología Económica y del Consumo, Universidad de La Frontera, Temuco 4811230, Chile; ligia.orellana@ufrontera.cl (L.O.); hector.poblete@ufrontera.cl (H.P.); k.beroiza01@ufromail.cl (K.B.); jose.sepulveda@ufrontera.cl (J.A.S.); k.reutter01@ufromail.cl (K.R.); 3Scientific and Technological Bioresource Nucleus (BIOREN-UFRO), Universidad de La Frontera, Temuco 4811230, Chile; 4Universidad Católica de Santiago de Guayaquil, Guayaquil 090615, Ecuador; maria.lapo@cu.ucsg.edu.ec; 5Centro de Investigación Escolar y Desarrollo, Cied-UCT, Universidad Católica de Temuco, Temuco 4780000, Chile; edgardo.miranda@gmx.com; 6Universidad Autónoma de Chile, Temuco 4801087, Chile; 7Departamento de Psicología, Universidad de La Frontera, Temuco 4811230, Chile; andres.concha@ufrontera.cl; 8Escuela de Trabajo Social, Universidad de Valparaíso, Valparaíso 2360102, Chile; mahia.saracostti@uv.cl; 9Departamento de Trabajo Social, Universidad de Chile, Santiago 8330015, Chile; 10Facultad de Economía y Negocios, Universidad de Talca, Talca 3465548, Chile; globos@utalca.cl; 11Departamento de Economía y Administración, Universidad Católica del Maule, Talca 3466706, Chile; cadasme@ucm.cl; 12Departamento de Trabajo Social, Universidad de La Frontera, Temuco 4811230, Chile; leonor.riquelme@ufrontera.cl

**Keywords:** family-to-work conflict, job satisfaction, psychological distress, dual-earner couples, dyadic analysis

## Abstract

The COVID-19 pandemic changed the work-family interface dynamics in some families. For couples who kept earning a double income during the pandemic, their family demands may entail a loss of psychological resources that affect the work domain. This study explored the intra-individual and inter-individual (crossover) direct and indirect effects of family-to-work conflict (FtoWC) on psychological distress and job satisfaction in a non-probabilistic sample of 860 different-sex dual-earner parents with adolescent children from Temuco and Rancagua, Chile. Mothers and fathers answered an online questionnaire measuring FtoWC, the Depression, Anxiety, and Stress Scale, and the Overall Job Satisfaction Scale. The data were analyzed using the actor-partner interdependence model with structural equation modeling. Results showed that a higher FtoWC is linked to greater psychological distress and lower job satisfaction in both parents. In contrast, psychological distress is directly linked to lower job satisfaction in fathers. In both fathers and mothers, they and their partners’ FtoWC were indirectly linked to lower job satisfaction via the fathers’ psychological distress. These findings indicate the need for gender-sensitive social and labor policies aimed at reducing the conflict between family and work to increase job satisfaction in both parents and reduce psychological distress, particularly in fathers.

## 1. Introduction

The COVID-19 health crisis has been challenging the work-family interface ever since March 2020 [1]. The pandemic has profoundly affected workers’ jobs and lives and heightened their levels of psychological distress (e.g., [2,3]), partly due to an increase in the conflict between the work and family domains [2,4]. According to Greenhaus & Beutell [5] (p. 77), “Work-family conflict is conceptualized as an inter-role conflict in which work and family appear to be mutually incompatible to some extent”. This incompatibility entails role strain or stress, as both domains compete for the limited resources—such as time and energy—that individuals invest in coping with everyday difficulties [6]. Work-to-family conflict (WtoFC) occurs when strain from the workplace negatively affects family life. In contrast, family-to-work conflict (FtoWC) occurs when the strain originating in the family domain affects the workplace [7,8]. 

Employees’ career uncertainty, workload, and job instability intensified during the pandemic, causing high stress and anxiety and leading to emotional exhaustion [9]. However, although WtoFC increased during the pandemic, several authors have highlighted the more significant increase of FtoWC [10]. The COVID-19 lockdown caused a crossover between work and family systems: parents were performing their duties as workers and children their duties as students while being compelled to share space (rooms) and possessions (laptops, tablets) [1]. Compared with routine family life before the pandemic, parents faced new and challenging demands and responsibilities in the family domain after the outbreak, such as household chores and childcare [11]. However, studies show that the pandemic reinforced traditional gender roles, as mothers spent more time in domestic activities than fathers [1].

Although evidence shows that teachers increased their readiness during the pandemic, significant changes in teaching methods negatively influenced student participation and learning [12]. Therefore, taking on the role of the educator was one of the parent’s biggest challenges during the pandemic, as emergency remote education implemented during the pandemic could not replace in-person classes successfully. This may help explain why FtoWC among parents increased symptoms of depression during the COVID-19 crisis but not WtoFC. Nevertheless, as working mothers also took on more responsibilities for homeschooling, they also reported a stronger impact on mental health symptoms than fathers [1]. Furthermore, workers’ FtoWC was coupled with feelings of guilt, mainly in working mothers, for not meeting job expectations [1] and thereby reducing employee job satisfaction [13]. 

Overall, these studies suggested that FtoWC, especially during the COVID-19 pandemic, may represent a unique source of stress for workers in couples and parents [2,4]. This stress can negatively impact not only one member of the couple’s mental health and job satisfaction, but it may cross over to also affect the mental health and job satisfaction of those close to them, such as their spouse [4]. Negative crossover effects between members of a dyad (e.g., a couple) refer to the process by which a stressor or strain (e.g., family demands) experienced by one member of the dyad affects the other member’s level of stress or strain [14]. The literature suggests that gender and gender equality are important in the crossover effects. While there are conflicting data about gender differences in crossover, more studies support unidirectional crossover from husbands to wives than from wives to husbands (e.g., [15]). Westman [14] (p. 145) indicates that “the crossover process is unidirectional, or at least stronger, from husbands to wives” because wives are more empathetic than husbands when faced with the stress or tension of their partners and because, as children, they are socialized to be caregivers and providers of support [14]. Unidirectional husband-to-wife crossover occurs more frequently in societies with greater gender inequality [15].

Most studies focused on the work-family interface during the pandemic have been conducted with samples of workers from the European Union, North America, and Asia [1]. They have also mainly focused on the individual level, neglecting the interdependency between the two members of a couple [1,4] and giving little to no consideration to the partner associations of parental FtoWC and spouses’ outcomes [1,11], considering gender differences. However, in mothers and fathers from dual-earner families, we recently discovered that job satisfaction mediates between FtoWC and family satisfaction, indicating that the connection between FtoWC and job satisfaction was negative in both parents [16]. Thus, this study focuses on the dynamics between FtoWC, psychological distress, and job satisfaction in different-sex dual-earner parents with adolescent children in a developing country in Latin America. In this region, culture and the still-present traditional gender roles may account for different work-family interface experiences and effects between fathers and mothers than those previously reported in developed countries. This approach fills in the gaps in the literature and adds new insights into the relationship between FtoWC and job satisfaction [16,17,18]. 

Chile is a developing country with a relatively collectivistic culture that places great importance on family, regardless of the parent’s gender, in dual-earner couples with adolescents [18]. Chile has many sociocultural factors that influence how families cope with the family-work conflict and the implications of that conflict. Traditional gender stereotypes persist, attributing to men attributes associated with leadership and competence while attributing to women characteristics associated with caring and sensitivity [19]. As a result of these stereotypes, women in Latin American countries have a greater presence in the informal labor sector and less capacity to address socioeconomic problems, and they are also primarily responsible for care work at home [20]. Furthermore, it is relevant for the purposes of this study to contextualize the quarantine system that Chile implemented to respond to the pandemic. Global health measures have included quarantines, border closures, social distancing, and the use of masks, among others. In particular, Chile’s 5-phase strategy (quarantine, transition, preparation, initial opening, and advanced opening) was implemented as a gradual measure according to the health situation of each commune in the country, establishing restrictions on people’s activities, mobility, and interaction [21,22]. These restrictions had a profound social, economic, and labor impact on the country, affecting working parents and, in particular, working mothers [23].

Families with adolescents were selected for this study because the literature shows that rearing adolescents tends to be more stressful than rearing younger children. For instance, both mothers and fathers with teenagers report being less happy than mothers and fathers with infants or toddlers while spending time with their children [24]. In addition, the demands of parent-adolescent dynamics increased during the pandemic as parents struggled to balance their work while supporting their adolescent children’s autonomy and educational and mental health needs [24]. Therefore, this study sought to explore the intraindividual and interindividual (crossover) direct and indirect effects of family-to-work conflict on psychological distress and job satisfaction in different-sex dual-earner parents with adolescents. The assessment of these effects follows the work-home resources model (W-HR) [6], the conservation of resource theory (COR) [25], boundary theory [26], and the Actor-Partner Interdependence Model (APIM) framework [27], which assesses how intra- and interindividual effects might be modified by features of the dyad members and their associations [28].

### Theoretical and Empirical Frameworks and Hypotheses

COR theory involves two major processes: “The first is a loss spiral, in which stress develops and resources further deplete, and the other is a gain spiral, in which resources accumulate” [6] (p. 547), [29]. Therefore, as COR theory posits, resources are critical to both creating and reducing stress and enhancing well-being in all areas [30]. In the COR theory, resources are defined as objects, personal characteristics, conditions, or energies that the individual values and which serve to meet their goals, cope with stress-generating demands, and acquire more resources. Individuals and organizations with greater resources are less likely to lose them and more able to gain them, while the opposite occurs with those lacking resources [25,29]. Several studies have used the COR theory to analyze pandemic-related resource gain or loss (e.g., [31,32,33]). Based on the COR theory, Hite and McDonald [31] investigated the urgent needs of the workforce and workplaces post-COVID-19, emphasizing the need to consider various strategies to assist employees in generating and retaining resources. Using the COR theory, Merino et al. [32] found that personal psychological resources, mainly vitality, were the most important predictors of eustress during lockdown in Spain, while job conditions, object resources (e.g., space availability, family interference with work), and energy (e.g., devoting time to work) predicted distress. In Thailand, Suthatorn and Charoensukmongkol [33] found that trust in organizations (an organizational resource) and trait mindfulness (a personal resource) mitigate the perceived stress of flight attendants. 

The link between FtoWC and job satisfaction is based on the COR theory and the W-HR model, which posits that conflict is a process whereby demands in one domain (family in the present study) deplete personal resources, resulting in diminished outcomes in the other domain (job in the present study) [6]. These processes can cross over between members of a couple [16]. Therefore, from the standpoint of the W-HR model, an individual dealing with FtoWC faces excessive demands from home, draining their limited personal resources that would otherwise be directed towards fulfilling roles and demands in the job [6]. Job satisfaction is the extent to which workers like their jobs [34] and is a key performance factor [7]. According to Venkatesh et al. [8], when individuals cannot meet the demands at work due to family interference, they perform less well and show less interest in their jobs. Family-related concerns, including problematic partnerships, children’s misbehavior [8], and family commitments, may prompt employees to allocate additional resources to the family sphere. This may decrease workers’ ability to perform their jobs [7], causing them to lose focus and realign schedules and priorities to meet their job requirements and family demands [7]. Indeed, employees who experience FtoWC will use their limited resources to solve their problems, leaving them with fewer resources to work, thus reducing their job satisfaction [13,35]. 

Nevertheless, there are conflicting data on the connection between FtoWC and job satisfaction. According to some studies, there is no strong association between FtoWC and job satisfaction (e.g., [36,37]). Contrary to what was proposed by the W-HR model, a study with dual-earner couples in Germany found that the outcomes of demands are primarily in the domain where the demand originates, i.e., only WtoFC was negatively associated with job satisfaction, whereas only FtoWC was negatively associated with family satisfaction [36]. Similarly, in a sample of Saudi female teachers, WtoFC was significantly and negatively related to job satisfaction, while the relationship between FtoWC and job satisfaction was not significant [37]. Other studies (including a recent meta-analysis) have reported a significant negative association between FtoWC and job satisfaction [13,16,38,39]. In particular, during the pandemic, FtoWC was associated with lower job performance [40] and job satisfaction in remote working conditions [41] because FtoWC creates negative consequences in the working environment [40]. For instance, Hong et al. [41] reported that work interruptions due to family demands are a strong antecedent of job satisfaction. These issues may be more marked in mothers in Chilean society, characterized by traditional gender stereotypes, where mothers are expected to prioritize their family role [19,20,21,22,23,24,25,26,27,28,29,30,31,32,33,34,35,36,37,38,39,40,41,42]. Regardless of the above, we propose the following hypothesis (intraindividual or actor effect, Figure 1):

**Hypothesis** **1.**
*Family-to-work conflict has a negative effect on job satisfaction for each parent.*


The COR theory posits that individuals obtain, retain, and protect things or resources they value to help them cope with stressors [25]. When measures to protect personal resources fail—for example; when they face high contextual expectations in the family domain—workers may experience psychological distress due to feeling unable to handle increased demands [43]. According to Yucel and Fan [44], when individuals experience FtoWC, they may overlook self-care practices that could avoid health issues due to time restrictions. Therefore, it follows that FtoWC increases general stress and ultimately results in psychological issues when excessive family demands and duties arise [44]. During the pandemic, FtoWC was linked to increases in psychological distress in workers in Canada and India [11] and depression in female social workers in China [43]. Although women in Chile reported a higher prevalence of distress than men [45], the gendered prevalence of mental health due to inequality in family and work responsibilities between men and women has also been reported in developed and developing countries before, during, and after the pandemic [1,46,47]. Thus, the following hypothesis addresses an important issue at a national and international level (intraindividual or actor effect, Figure 1): 

**Hypothesis** **2.**
*Family-to-work conflict has a positive effect on psychological distress for each parent.*


The COR theory also involves a loss spiral, in which distress symptoms threaten or deplete personal resources (e.g., energy, skills [25]), which may negatively affect job performance and lead to a decline in job satisfaction. In this regard, the evidence shows that psychological distress reduces the employees’ ability to cope with job demands, which may severely reduce their job performance. At the same time, distress has been identified as a major cause of low job satisfaction in both men and women in developed and developing countries [48,49]. Thus, the following hypothesis is proposed for diverse contexts (intraindividual or actor effect, Figure 1): 

**Hypothesis** **3.**
*Psychological distress has a negative effect on job satisfaction for each parent.*


The FtoWC might also have crossover effects [14], but the empirical evidence is sparse and mixed. In a recent meta-analysis, Li et al. [50] discovered a negative correlation between one partner’s FtoWC and the other partner’s work attitudes, including job satisfaction and engagement. This study shows that, regardless of gender, family role conflicts affect individuals and their partners; however, other findings point to an asymmetrical transmission or strain in different-sex couples, i.e., one partner influences the other but not the other way around. During the pandemic in Chile, we found that only the mothers’ FtoWC crossed over to the fathers’ job satisfaction, negatively impacting the fathers’ well-being, but not vice versa [16]. There is also evidence of a lack of crossover effects between one member of a couple’s FtoWC and the other’s job satisfaction [36]. Thus, we posited the following hypothesis regarding the interindividual effect (crossover or partner effect):

**Hypothesis** **4.**
*Family-to-work conflict of one parent has a negative effect on the other parent’s job satisfaction.*


The COR theory also posits [51] “that the threat of loss of a resource in one partner constitutes a further resource threat to the other partner, which can in turn worsen his or her response to his or her own threats of resource loss”. This suggests that FtoWC, as a resource loss, may not only affect the subject’s health but can also cross over to their partner and impair their health. Despite the lack of evidence of crossover effects from one member of the couple’s FtoWC on the other’s mental health [51], some pre-pandemic studies support the notion. In Germany, Yucel and Latshaw [36] reported that in dual-earner couples, both members’ FtoWC was related to a decline in their mental health, and both men’s and women’s mental health suffered if their partners’ FtoWC increased (crossover effects). During the pandemic in China, however, using a sample of dual-earner parents with at least one child aged 7–18 years old, Zou et al. [11] found no crossover effects between spouses despite both parents’ FtoWC being strongly connected to their own symptoms of depression. Thus, the next hypothesis posited that: 

**Hypothesis** **5.**
*Family-to-work conflict of one parent has a positive effect on the other parent’s psychological distress.*


To the best of the authors’ knowledge, there are no available studies that have evaluated the crossover effects between one member of a couple’s psychological distress and the other member’s job satisfaction; however, there is evidence showing that individuals’ mental health can affect their partners’ well-being [17,52,53,54,55]. Schnettler et al. [52] found that a man’s depression was negatively associated with a woman’s satisfaction with family life and vice versa. Similar results were reported by Orellana et al. [17] when assessing partner effects between psychological distress and satisfaction with family life in dual-earner parents. However, there is also evidence of asymmetrical crossover effects from men to women. Chi et al. [53] found that psychological distress is transmitted only from fathers to mothers. Moreover, in a longitudinal study, Guo [54] found that a husband’s depressive symptoms crossed over to his wife’s life satisfaction, negatively affecting their well-being two years later. We argue that similar results may be expected for psychological distress and job satisfaction and thus posit the following hypothesis (Figure 1): 

**Hypothesis** **6.***Psychological distress of one parent has a negative effect on the other parent’s job satisfaction*.

Nevertheless, we expected more significant unidirectional crossover effects from men to women given Chile’s male-dominated culture and female socialization [14], as this occurs more frequently in societies with greater gender disparity [15].

Bagger and Li [39] claim that interference from family members makes fulfilling job obligations more difficult and increases psychological stress. Greater psychological stress brought on by FtoWC may result in a person receiving fewer positive performance evaluations because people absorb and interpret information in ways that reflect their emotional states. In parallel, Venkatesh et al. [8] reported that individuals who experience high levels of FtoWC are more likely to feel overwhelmed by their failure to meet the demands at work and thus experience a depletion of mental resources to cope with their work. Combining the aforementioned studies, it is reasonable to suggest that psychological distress may be a link between FtoWC and job satisfaction. 

The boundary theory explains the mutual influence between the work and family domains, describing how individuals create and maintain boundaries between work and family [26]. This theory seeks to explain how individuals navigate the overlap between work and family by segmenting or integrating these domains and the consequences of blurred versus sharpened boundaries between them [56]. Blurred boundaries may have been heightened during the pandemic, as boundary violations from family to work were more likely to occur due to non-existent physical and temporal boundaries [1]. For instance, it may be possible that during the pandemic, parents had to spend more time at work to fulfill task requirements because children interrupted their work by asking for help with their schoolwork, thus increasing parents’ psychological distress [1]. In addition, the boundary between work and personal life presented additional problems during the pandemic, potentially resulting in conflict between family and work due to changes in the workplace and home environment [40], i.e., sharing space and computers or tablets so that parents could fulfill their work roles and children their student roles [1]. In this regard, Rasool et al. [57] discovered that psychological distress brought on by a negative work environment lowers employees’ productivity. During the pandemic, psychological distress brought on by a lack of boundaries between work and home may have contributed to this poor working environment. Therefore, as work and family boundaries overlap and both systems could be more integrated [56], psychological distress caused by FtoWC may affect the workers’ on-the-job performance and thus decrease their job satisfaction. On this basis, we propose that psychological distress may mediate between FtoWC and job satisfaction.

In this regard, there is evidence of psychological distress acting as an intraindividual mediator between elements of the work-family interface and an individual’s well-being, i.e., between work-to-family enrichment and satisfaction with family life in men [17]. There is also evidence of a possible interindividual mediating role for psychological distress. Ten Brummelhuis et al. [55] found that women’s distress mediates between their own energy deficit due to work and family demands and their spouses’ distress. Therefore, the following hypothesis was posited (Figure 1):

**Hypothesis** **7.**
*Both parents' psychological distress mediates between their family-to-work conflict and job satisfaction (actor and partner effects).*


Family-related stressors can lead to FtoWC in both parents (e.g., children’s misbehavior), but mothers may be more commonly responsible for managing them [8]. During the pandemic, both women and men experienced frustration with managing their workloads; however, men tended to adopt the “assistant” role or perform poorly. Therefore, while men’s paid work was given priority [1], women were more responsible for unpaid childcare, schooling, and domestic tasks [1,16]. It appears that the COVID-19 pandemic forced both men and women to fill conventional roles [1]. Hence, women being overburdened with caregiving and the overall workload were related to higher FtoWC, parenting stress, stress, depression, and anxiety symptoms than in men (e.g., [1,43]). 

Historically, in Chile, women have faced more difficulties than men in entering the labor market due to structural factors, including family responsibilities and access to education [58]. Nevertheless, despite Chilean female participation in the workforce being below that of Latin America and developed countries, women’s labor participation rate has improved nationally by three percentage points in seven years, reaching 53 percent at the end of 2019 [59]. In 2012, Standard 3262 (NCh 3262:2012) was founded in Chile to encourage gender equality in organizations, suggesting strategies to produce a culture shift to narrow the gender gap by promoting work-family life balance and co-parenting [60]. However, as of 2022, only 41 private and public Chilean organizations had been certified by this standard [60]. Despite efforts to narrow gender gaps, gender disparities in paid employment and job quality persist in Chile (e.g., see [61]). According to the OECD Report 2022, Chilean women earn 21.7% less than men, a gap that drops to 18.8% in formal jobs and rises to 30.1% in informal jobs [23]. 

The women’s labor participation rate in Chile fell to 43.9% in 2020 due to the global COVID-19 pandemic [59]. This percentage represents a population whose well-being is constantly made more difficult by work and family obligations. Patriarchal demands hamper women’s labor development in Latin America because women, especially mothers, are expected to prioritize their family role [42]. Women, for example, devote 3.2 h more per day than men to domestic or unpaid work [62]. Chile’s position in the Global Gender Gap Index Ranking [61] reflects this gender disparity, ranking 57th out of 153 countries and 14th out of 25 Latin American and Caribbean countries. Working women in Chile also report a high-to-very-high prevalence of distress (37%, [45]), and differences in men’s and women’s family and work responsibilities contribute to the gendered prevalence of mental health issues [46].

For these reasons, in an exploratory manner (i.e., no hypothesis), we also suggest that, particularly during the pandemic, distinct gender patterns may be observed in inter- and intraindividual effects between FtoWC, psychological distress, and job satisfaction.

Therefore, in this study, we hypothesized that both parents’ family-to-work conflict positively affects their own psychological distress and negatively affects their own job satisfaction, and that both parents’ psychological distress negatively affects their own job satisfaction (intraindividual effects). At the same time, we hypothesized that one parent’s family-to-work conflict positively affects psychological distress and negatively affects the other parent’s job satisfaction, while we also expect that one parent’s psychological distress will affect the other parent’s job satisfaction (interindividual or crossover effects). We also hypothesized that both parents’ psychological distress may mediate the effects of family-to-work conflict on job satisfaction at intra- and interindividual levels. Nevertheless, in light of the evidence that the pandemic compelled both men and women to assume traditional gender roles [1] and the cultural norms in Chile, we also investigated the various gender patterns in the hypothesized relationships between psychological distress, job satisfaction, and family-to-work conflict.

## 2. Materials and Methods

This study is part of a longitudinal study exploring how job, family, and food domains are intertwined in Chilean families. Using non-probability sampling, 860 different-sex dual-earner parents (married or cohabiting) with at least one child aged between 10 and 15 years old were recruited in Temuco and Rancagua in the Araucanía and O’Higgins regions of Chile, respectively (Table 1). The results of the latest Chilean National Census in 2017 indicate that in both regions, the average schooling age is 9 years in the population aged 25 or over. In both regions, the largest proportion of the labor-active population works in the tertiary sector (over 70% in O’Higgins and over 80% in La Araucanía); however, the O’Higgins Region stands out for having a higher proportion of the population that works in the primary sector (22% versus 13% in La Araucanía). Approximately 70% of the population in both regions resides in urban areas.

Parents were invited to participate in this study via their children’s schools, which serve socioeconomically diverse backgrounds. They were informed by trained interviewers of this study’s objectives, questionnaire structure, and the anonymity and confidentiality of their responses. Participating families were contacted by a professional interviewer, who liaised with one family member by phone and email (often the mother). Interviewers emailed the links to the three surveys (one for each family member) to this family member between March and July 2020 in Rancagua and August and December 2020 in Temuco. After the COVID-19 pandemic was declared by mid-March 2020 in Chile, Rancagua and Temuco were on mandatory lockdown in June and July 2020 and November and December 2020, respectively. However, workers from various sectors worked remotely in March and throughout the year. Once the interviews were concluded, each participating couple received a gift card worth approximately 15 USD. 

In this study’s pilot test, which used the same recruiting strategy and data collection process as previously mentioned, 50 families in Temuco participated. The pilot test showed satisfactory results; therefore, no changes were made to the instrument or the data collection procedure. The Universidad de La Frontera Ethics Committee approved this study (protocol 007/19).

### 2.1. Measures

The family-to-work conflict was measured with four items, adapted by Kinnunen et al. [63]. These items ask about negative spillover from family to work (example item: “The demands of your family or spouse/partner interfere with your work-related activities?”). Response options are presented on a five-point scale (1: never; 5: very often). This test was administered using the verified Spanish version [16]. Family-to-work conflict scores were obtained by adding the scores from the four items, with higher scores representing a higher family-to-work conflict (score range 5–20). In the present study, the standardized factor loadings ranged from 0.854 to 0.898 for mothers and from 0.881 to 0.935 for fathers, all statistically significant (*p* < 0.001). The AVE values were higher than 0.50 (AVE mothers = 0.78, fathers = 0.81). This measure also showed good internal reliability, with an Omega coefficient of 0.93 for mothers and 0.95 for fathers. 

Psychological distress was measured with the Depression, Anxiety, and Stress Scale (DASS-21) [64]. This scale has 21 items grouped in three subscales measuring subclinical depression (e.g., “I could not seem to experience any positive feeling”), stress (e.g., “I experienced breathing difficulty”), and anxiety (e.g., “I found it hard to wind down”). The participants used a four-point severity scale to rate the frequency with which they had experienced each state during the week (0: it does not describe anything that happened to me or that I felt during the week; 3: yes, this happened to me a lot, or almost always). The discriminant validity analysis in this study showed that all 21 items were in one dimension instead of three. Osman et al. [65] reported a similar finding, showing that the general factor of psychological distress explained a more significant proportion of the sample variance than the individual dimensions and that most of the DASS-21 items did not contribute solely to the dimension specifically presented. Therefore, the 21 items in this study were grouped into a single dimension called psychological distress (DASS in the results). Orellana et al. validated this scale in Chilean adults [17]. DASS-21 scores were obtained by totaling the scores from the 21 items, with higher scores representing higher psychological distress (score range 0–63). This study’s standardized factor loadings were significant (*p* < 0.001), ranging from 0.669 to 0.916 for mothers and 0.745 to 0.925 for fathers. The AVE values were higher than 0.50 (mothers’ AVE = 0.67, fathers’ AVE = 0.68). Moreover, a good reliability index was obtained (mothers’ omega = 0.98; fathers’ omega = 0.98).

Job satisfaction was measured with the Overall Job Satisfaction Scale (OJSS) [34], which comprises six items (example item: “Most days I am enthusiastic about my job”). Respondents indicated their degree of agreement with each statement using a five-point Likert scale (1: completely disagree; 5: completely agree). The Spanish version of the OJSS scale was used, which has been validated and has shown good internal consistency in studies with couples in Chile [16,18]. OJSS scores were obtained by totaling the scores from the six items, with higher scores representing higher job satisfaction (score range 6–30). In the present study, the standardized factor loadings of the OJSS scale ranged from 0.567 to 0.939 for mothers and from 0.539 to 0.910 for fathers, all statistically significant (*p* < 0.001). The AVE values were higher than 0.50 (AVE mothers = 0.68, fathers = 0.65). The OJJS showed good internal reliability, with Omega coefficients of 0.93 for mothers and 0.92 for fathers.

Both parents also disclosed their age, type of employment, number of weekly working hours, and monthly income. Mothers reported the number of family members and children in the household. The combination of monthly household income ranges and the number of family members in a matrix determines socioeconomic status (SES) [66].

### 2.2. Data Analysis

Descriptive analyses were conducted using the Statistical Package of Social Sciences (SPSS) v. 23. To test Hypotheses 1–6, the Actor-Partner Interdependence Model (APIM) was used, in which the unit of analysis for the APIM is the dyadic interaction [27]. The APIM is the most frequently used analytical model of dyadic data because it can model the interdependence between dyadic members [67]. The APIM allows simultaneous estimates of the effects of an individual’s attributes on their own outcome variable (the actor effect, intraindividual) and their partner’s (the partner effect, interindividual, crossover). Partner effects are part of the interdependent nature of dyadic relationships [28]. In this study, the APIM with distinguishable dyads (each member should be theoretically distinguishable, i.e., mother-father dyads) was tested using structural equation modeling (SEM) [27] with Mplus 8.4. 

In this study, fathers and mothers participated in the analysis as both actors and partners. APIM associations between variables for the same parent are “actor effects” (intraindividual), and associations between variables from one parent to the other are “partner effects” (interindividual). Actor effects associate each parent’s family-to-work conflict (FtoWC), psychological distress (DASS), and job satisfaction (OJSS), as well as each parent’s DASS and OJSS. Partner effects relate one parent’s FtoWC to the other‘s DASS and OJSS, and one parent’s DASS to the other’s OJSS. 

In addition to actor and partner effects, the APIM controls for other effects. In this regard, a correlation between the independent variables provided by each parent was specified to account for one parent’s influence on the other in terms of FtoWC interdependency. Other sources of interdependence between partners were controlled for, as suggested by Kenny et al. [27], by specifying correlations between the residual errors of the dependent variables (DASS and OJSS) of the two parents. 

Other effects controlled for were the parents’ age, type of employment and number of working hours, family SES, number of children, and city of residence. These variables were thus incorporated into the model with direct effects on the dependent variables of both parents (DASS and OJSS).

The SEM was conducted using the weighted least squares mean and variance adjusted (WLSMV) to estimate the structural model parameters. The polychoric correlation matrix was considered for the SEM analysis because the items were ordinal. The model fit of the data were determined using the following values: the Tucker-Lewis index (TLI) and the comparative fit index (CFI), which show a good fit with values above 0.95, and the root mean square error of approximation (RMSEA), which shows a good fit when values are below 0.06 [68]. 

Lastly, to test Hypothesis 7, an SEM through a bias-corrected (BC) bootstrap confidence interval using 1000 samples [69] was conducted to test the mediating role of psychological distress between the independent and dependent variables. A mediating role is supported when BC confidence intervals do not include zero.

## 3. Results

### 3.1. Sample Description

A total of 860 dual-earner couples with at least one adolescent child aged 10–16 participated in this study, resulting in 860 responses from mothers and fathers. The sociodemographic characteristics of the sample are displayed in Table 1. The mean age for fathers was 42.2 years, and for mothers, 39.0 years. On average, the family size was four family members, including two children. Most families belonged to the middle SES. Most participants were employees and worked full-time (45 h per week). Fathers were employed (versus self-employed) with a full-time job more than mothers. Regarding differences between cities, Temuco had a greater proportion of families of low SES, and Rancagua had a greater presence of families of middle SES (*p* < 0.001). 

Table 2 shows the average scores and correlations for family-to-work conflict (FtoWC), psychological distress (DASS), and job satisfaction (OJSS). All the correlations were significant and in the expected direction. Mothers scored significantly higher than fathers in FtoWC (t = 5.324, *p* < 0.001) and in DASS (t = 7.107, *p* < 0.001). Mothers and fathers did not differ in the average scores for OJSS (t = 0.934, *p* = 0.351). 

### 3.2. APIM Results: Testing Actor-Partner Hypotheses

The results from the estimation of the structural model are shown in Figure 2. The model that assessed the APIM association between the mother’s and father’s family-to-work conflict (FtoWC), psychological distress (DASS), and job satisfaction (OJSS) fitted the data well (CFI = 0.983; TLI = 0.982; RMSEA = 0.024). A significant correlation (covariance) was found between the two parents’ FtoWC (r = 0.482, *p* < 0.001) and between the residual errors of the mother’s and father’s DASS (r = 0.397, *p* < 0.001) and OJSS (r = 0.229, *p* < 0.001).

Hypothesis 1 (H1) states that family-to-work conflict has a negative effect on job satisfaction for each parent. The path coefficients (standardized) indicated that the fathers’ (γ = −0.092, *p* = 0.029) and mothers’ (γ = −0.176, *p* < 0.001) FtoWC had a negative effect on their own OJSS (H1 supported). Hypothesis 2 (H2) states that family-to-work conflict has a positive effect on the psychological distress of each parent. The fathers’ (γ = 0.275, *p* < 0.001) and mothers’ (γ = 0.369, *p* < 0.001) FtoWC positively affected their own DASS (H2 supported). Hypothesis 3 (H3) states that psychological distress has a negative effect on job satisfaction for each parent. While the fathers’ DASS negatively affected their OJSS (γ = −0.257, *p* < 0.001), the mothers’ DASS had no significant effect on their OJSS (γ = −0.045, *p* = 0.284). These findings supported H3 only for fathers (Figure 2).

The next three hypotheses seek partner effects. Hypothesis 4 (H4) states that the family-to-work conflict of one parent has a negative effect on the other’s job satisfaction. The fathers’ FtoWC did not significantly affect the mothers’ OJSS (γ = −0.006, *p* = 0.884); likewise, the mother’s FtoWC did not significantly affect the fathers’ OJSS (γ = −0.035, *p* = 0.441). These findings did not support H4. Hypothesis 5 (H5) stated that the family-to-work conflict of one parent has a positive effect on the other parent’s psychological distress. The fathers’ FtoWC did not significantly affect the mothers’ DASS (γ = 0.043, *p* = 0.326), while the mothers’ FtoWC had a significantly positive effect on the fathers’ DASS (γ = −0.188, *p* < 0.001), thus supporting H5 only for mothers. Hypothesis 6 (H6) states that the psychological distress of one parent has a negative effect on the other parent’s job satisfaction. While the fathers’ DASS negatively affected the mother’s OJSS (γ = −0.116, *p* = 0.021), the mothers’ DASS did not have a significant effect on the fathers’ OJSS (γ = −0.031, *p* = 0.481); therefore, H6 was supported only for fathers. 

Most control variables did not significantly affect the model (Table A1 in Appendix A). The family’s SES negatively affected the mother’s DASS (γ = −0.101, *p* < 0.05) and positively affected her OJSS (γ = 0.105, *p* < 0.01). Mother’s age positively affected her own OJSS (γ = 0.105, *p* < 0.01). At the same time, a similar result was found for father’s age and OJSS (γ = 0.091, *p* < 0.05). The mothers’ type of employment (employed vs. self-employed) also positively affected their own job satisfaction (γ = 0.081, *p* < 0.05), i.e., self-employed mothers reported higher job satisfaction than employed mothers.

### 3.3. Testing the Mediating Role of Psychological Distress

Lastly, this study tested the mediating role of both parents’ psychological distress between the two parents’ family-to-work conflict and job satisfaction. The role of the fathers’ DASS as a mediator in the relationship between their own FtoWC and OJSS was supported by a significant indirect effect obtained with the bootstrapping confidence interval procedure (standardized indirect effect = −0.071, 95% CI = −0.091, −0.031, Table 3). The role of the fathers’ DASS as a mediator in the relationship between the mothers’ FtoWC and OJSS was also supported (standardized indirect effect = −0.032, 95% CI = −0.041, −0.002). Similar results were found for the role of the fathers’ DASS as a mediator in the relationship between the mothers’ FtoWC and fathers’ OJSS (standardized indirect effect = −0.048, 95% CI = −0.075, −0.018). In addition, the role of the fathers’ DASS as a mediator in the relationship between fathers’ FtoWC and mothers’ OJSS was supported (standardized indirect effect = −0.028, 95% CI = −0.054, −0.002).

No other indirect effects of the parents’ psychological distress were found, as the confidence intervals included zero (Table 3). These findings partially supported Hypothesis 7 regarding the mediating role of psychological distress between parents’ FtoWC and job satisfaction. 

## 4. Discussion

Following the W-HR model [6], the COR theory [25], and the boundary theory [26], we proposed that the high demands workers experience in their family domain can cause a loss spiral of personal resources that affects their own (intraindividual) and their partners’ (interindividual) psychological distress and job satisfaction. Results of this study showed that, for different-sex dual-earner parents with adolescents, a higher family-to-work conflict is linked to higher psychological distress and lower job satisfaction, regardless of the parent’s gender. By contrast, only the father's psychological distress directly reduced his job satisfaction. Our results also show asymmetrical interindividual or crossover effects: the mothers’ family-to-work conflict negatively affected the fathers’ psychological distress, and the fathers’ psychological distress negatively affected the mothers’ job satisfaction. Furthermore, our findings show that only fathers’ psychological distress mediates between parents’ family-to-work conflict and job satisfaction. These findings are discussed in detail below.

### 4.1. Actor Effects

We expected that a higher resource loss derived from FtoWC would reduce job satisfaction. Consistent with the W-HR model [6], FtoWC conflict had a negative effect on job satisfaction for mothers and fathers (H1 supported), confirming previous findings reported in studies carried out before and during the pandemic [13,16,38,39]. These findings align with the idea that when individuals cannot meet the demands at work due to family interference with their job [8], employees invest more resources in the family domain. In contrast, resource consumption means employees have fewer resources, decreasing their ability to perform their job [7,8], thus reducing their job satisfaction [13,35]. Our findings are also in line with the suggestions by Hong et al. [41] regarding work interruptions due to family demands, which seem to be a strong antecedent of job satisfaction, highlighting the importance of considering non-work-related factors in job satisfaction. Contrary to what was expected for the Chilean context regarding traditional gender stereotypes, the paths between family-to-work conflict and job satisfaction were of low strength for mothers and fathers, which is in line with studies reporting that the cross-domain relationship between work and family is of low strength [e.g., [38]]. However, it should be noted that these paths were not as strong as the ones we found in a previous study conducted in Chile [16], which may be related to the different samples used in both studies. Another possible explanation is that the direct effect of family-to-work conflict on job satisfaction is lower when psychological distress is included in the APIM model, underscoring the importance of assessing psychological distress as a mediating variable between the two constructs.

Additionally, we anticipated that higher FtoWC would be linked to greater psychological distress for each parent. This hypothesis (H2) was supported by the COR theory [43] and is consistent with the growing body of research showing that excessive family obligations are linked to mental health issues, like the psychological distress experienced during the pandemic [43,44]. In this regard, social distancing and lockdowns implemented to stem the spread of COVID-19 have reportedly drastically reduced several contextual resources (e.g., lack of support for household chores, the reduction or closing of childcare services and schools; [2]) that people can typically turn to for help in dealing with psychosocial stressors, which may be associated with increased stress, anxiety, and depression [3].

Despite being significant for both mothers and fathers, the negative effect of FtoWC on psychological distress was of medium strength for mothers and low strength for fathers. These findings are in line with other investigations, both before and during the pandemic, which show that mothers report considerably higher levels of FtoWC and psychological distress than fathers do [1,8,43]. In terms of the pandemic, studies conducted during lockdowns at the beginning of the pandemic revealed that women experienced a significant increase in childcare and household tasks than men did [1,16], with the men assuming the role of “assistant” or performing poorly [1], which may imply an increased burden for mothers. Contrary to what was expected, this result is also consistent with Chilean culture, characterized by traditional gender stereotypes and the expectation that mothers will prioritize their family role, although they have a paid job [19,42]. Thus, gender differences in the strength of the path from FtoWC to psychological distress may be higher in countries and cultures with a wider gender gap.

As indicated by Hypothesis H3, we expected that higher psychological distress would have a negative effect on job satisfaction for fathers and mothers, respectively. However, our results show that psychological distress negatively affected job satisfaction in fathers, not mothers (H3 partially supported). Therefore, our findings for fathers are in line with the loss spiral of resources posited by the COR theory [25] and with evidence showing that psychological distress reduces the employees’ ability to cope with job demands, which, in turn, may decrease their job performance [48], negatively affecting their job satisfaction [38,48,49]. Nevertheless, all the studies above evaluated the association between psychological distress and job satisfaction, considering work-related stressors. Therefore, our results expand on the knowledge showing that family-related stressors may also reduce job satisfaction, at least in the case of fathers during the pandemic.

Regarding the null effect in mothers, this result implies gender differences, which could be related to different degrees of integration of the work and family domains, according to boundary theory [26]. For mothers, family conflict does not appear to deplete psychological resources that can be invested in their job satisfaction. The opposite trend was found in fathers. As a result, while there is evidence that both men and women encountered poorly defined boundaries between work and family during the pandemic, with mothers being particularly affected [1], it is possible to hypothesize that fathers adopted a more integrated interaction between work and family. In contrast, mothers adopted a more segmented interaction between the two domains, so their psychological distress did not pass over to work, and thus their job satisfaction was not affected. In addition, it is also possible that women may have a greater capacity to segment their family and work lives, given that mothers may be more frequently tasked with dealing with family stressors than fathers [8]. However, our results confirm the importance of considering a gendered perspective on boundary theory [16,56]. Contrary to expectations that these results would be similar across societies and cultures, an alternative explanation may be related to the traditional gender roles that still prevail in Chile and which were exacerbated during the pandemic [1], where family roles are primarily for women and work roles are secondary, whereas men’s work role outside the home is considered to be providing for the family [16,43]. Thus, although mothers experienced more psychological distress and FtoWC than fathers, it is possible that their job satisfaction was not affected by their psychological distress because they assumed their gender role and prioritized their family over their work, while men felt their gender role was threatened by the psychological distress they faced because of the FtoWC, probably because work is fundamental to men’s identity [49].

### 4.2. Partner Effects

This study also tested crossover effects between these dual-earner parents (H4–H6). It was proposed that workers in dual-earner couples experiencing higher family-to-work conflict would mutually affect their job satisfaction [14]. Contrary to expectations [17,50], the resource loss by one parent facing high family demands did not cross over to the other parent’s job satisfaction (H4 not supported). This result may be explained considering that when actor effects are weak, as in the present study, it is likely that partner effects do not exist [14], as was previously reported in dyadic studies with dual-earner couples [e.g., [17,18]]. However, our findings are consistent with results reported by Yucel and Latshaw [36], which may be related to the fact that partner effects may be indirect effects mediated by other variables [14]. This study may be a case of the fathers’ psychological distress, which will be discussed below.

We also expected that one parent’s FtoWC would positively affect the other’s psychological distress (Hypothesis 5). This hypothesis was partially supported. The mothers’ FtoWC positively crossed over to the fathers’ psychological distress, but not vice versa. Thus, our findings partially support that FtoWC, as resource losses, may not only affect the individual’s health but can cross over to their partner and impair their health [25,51]. Among the three possible mechanisms of stress transmission across members of a couple proposed by Westman [14], our findings are consistent with the mechanism that contends that because partners’ lives are intertwined, the stresses experienced by one are frequently felt by the other simply by shared experiences, homes, and children, as in the sample here, data from which were collected during the COVID-19 pandemic when all family members were generally required to remain at home full time [2]. 

Thus, FtoWC can potentially heighten existing conflicts and/or increase additional conflicts, which in turn can have a negative impact on individuals’ mental health. For instance, in Canada, Chai and Schieman [70] found that FtoWC intensifies the negative effect of perceived housework unfairness on relationship quality and the negative effect of spousal disputes on relationship quality, regardless of gender. The latter may clarify the negative crossover effect from the mothers’ FtoWC to the fathers’ psychological distress, but not the absence of a crossover effect in the opposite direction. Based on earlier research conducted in Germany that discovered symmetrical crossover effects between men’s and women’s FtoWC and mental health, we anticipated a bidirectional or symmetrical crossover [36]. The discrepancies between studies conducted in Germany and Chile may reflect gender inequalities associated with the culture or country. Of the 153 countries in the Global Gender Gap Index Ranking, Germany ranks in 10th place, while Chile ranks 57th [51]. According to gender role theory and traditional gendered socialization in Latin American nations, women are expected to care for their homes and families regardless of their outside employment status [18]. In contrast, men’s work role outside the home is considered providing for the family [16]. Thus, in this cultural context, women bore greater responsibility for childcare, schooling, and household tasks during the pandemic [18], which resulted in higher FtoWC in mothers than in fathers. Thus, it is possible to hypothesize that the higher resource loss in mothers may lead to them being unable to assume all the household chores, meaning the father must become more involved in the home to compensate for the mothers’ conflict. This, in turn, may negatively affect the fathers’ psychological distress. It is important to remember that fathers’ mental health negatively impacts both their own and their mothers’ FtoWC.

We also expected that the psychological distress of one parent would have a negative effect on the other parent’s job satisfaction. Hypothesis 6 was also partially supported: the fathers’ psychological distress negatively crossed over to the mothers’ job satisfaction, but not vice versa. Previous findings showing symmetrical crossover effects between the two members of a couple’s mental health problems and satisfaction with family life are refuted by this asymmetrical crossover [17,18], suggesting gender differences according to the domain that receives the partners’ strain. 

According to the COR theory [6,25], the asymmetrical crossover effect from the fathers’ psychological distress to the mothers’ job satisfaction may reflect a dynamic of a resource loss spiral in which fathers reduce their availability of resources to provide social, emotional, or instrumental support to mothers, which in turn may reduce the mothers’ job performance and satisfaction [7,13]. Thus, our findings expand on this knowledge, showing that a loss of psychological resources due to FtoWC in fathers affects their job satisfaction and the mothers’. Nevertheless, this reasoning may explain the negative crossover effect from the mothers’ FtoWC to the fathers’ psychological distress, but not the lack of a crossover effect the other way around.

However, our findings align with a previous study showing an asymmetrical crossover from men’s depressive symptoms to their wives’ life satisfaction [54]. One feasible explanation of our findings may be related to the results of Chi et al. [53] in a longitudinal study with father-mother-adolescent triads. These authors found a stable transmission of psychological distress only from fathers to mothers, possibly due to the hierarchy of emotional influence in families, in which fathers could be more influential than other family members [53]. This result is consistent with the traditional family structure in Latin American countries like Chile, where fathers are usually the main breadwinners, and at home, they still occupy a higher status than mothers within the family, although mothers have a paid job [18]. Furthermore, these findings may also reflect women’s traditional socialization, which encourages them to be more empathetic and sensitive to the moods and problems of their male partners [18,23].

It should be noted that although Chilean society is characterized by gender inequality, our results show not only a unidirectional crossover from fathers to mothers (psychological distress to job satisfaction) but also a unidirectional crossover from mothers to fathers (family-to-work conflict to psychological distress). These results partly contradict suggestions that unidirectional crossovers are more common from men to women [14] and in societies with greater gender inequality [15]. As these results may be a consequence of the pandemic's impacts on the work-family interface, further research is needed post-pandemic.

### 4.3. The Mediating Role of Psychological Distress

In this study, psychological distress mediated between family-to-work conflict and job satisfaction at both the intra- and interindividual levels, but only via the fathers’ psychological distress. In other words, the fathers’ psychological distress serves as a conduit for the stress that results from their family-to-work conflict, being transferred indirectly to their job satisfaction, as it seems that both parents’ FtoWC drains the fathers’ psychological resources, which not only prevents them from investing resources in their own job but also the mothers’. 

At an intraindividual level, the indirect effect of FtoWC on job satisfaction in fathers via their own psychological distress aligns with earlier studies by Bagger and Li [39] and Venkatesh et al. [8], while this finding is also in line with a previous study showing that psychological distress mediates the relationship between work-to-family enrichment and family satisfaction in fathers [17]. However, our findings expand on this knowledge, showing that psychological distress in men may mediate not only the associations between resources acquired in the work domain but also between a resource loss originating in the family domain, affecting not only their family life satisfaction but also their job satisfaction. 

In keeping with a previous result, the fathers’ psychological distress mediates between their own FtoWC and the mothers’ job satisfaction. This result validates our earlier crossover hypothesis, indicating that the fathers’ FtoWC indirectly negatively affected the mothers’ job satisfaction by first negatively affecting the fathers’ psychological distress. Mothers’ negative emotional responses to the fathers’ psychological stress and other negative responses to their own conflict may corroborate this connection [14], as fathers with FtoWC allocate resources once intended for their job to their family [e.g., [16]]. This is an important finding because although the mothers’ psychological distress originated in their own FtoWC and did not directly affect their job satisfaction, it was negatively affected by the fathers’ FtoWC via their psychological distress. 

Both parents also experience a second negative indirect effect on their job satisfaction, from the mothers’ FtoWC via the fathers’ psychological distress. The fathers’ psychological distress mediates between the mothers’ FtoWC and both parents’ job satisfaction. These findings may be partly explained by the higher FtoWC experienced by mothers during the pandemic [43], which negatively affected their job satisfaction and their partners’ indirectly. Although mothers in our sample scored higher than fathers in the DASS-21, these findings suggest that the fathers’ mental health may be more sensitive to the mothers’ FtoWC than vice versa and that the fathers’ psychological distress has a greater potential to be transmitted to their partners, negatively affecting their well-being. 

Although these results confirm the interindividual mediating role of distress reported in a pre-pandemic study (via women’s distress; [55]), our results underscore the relevance of the fathers’ psychological distress during the pandemic as one underlying mechanism that explains the effects of both parents’ higher family demands on their well-being in the work domain. One possible explanation may be that boundary violations from family to work and the related issues [1,43] generate a negative workplace environment that exacerbates the worker’s psychological distress, which in turn may negatively affect their productivity and job satisfaction. However, this condition may have a stronger impact on fathers due to their traditional gender role as the family’s provider [1,43]. These results reinforce the idea that, although involuntary [1], fathers adopted a more integrated interaction between work and family during the pandemic, affecting both their job satisfaction and the mothers’.

Regarding the significant effects of some of the control variables used in the APIM model, the positive effect of the family’s SES on the mothers’ OJSS and the negative effect of the family’s SES on the mothers’ DASS align with research reporting that employees with lower SES have lower job satisfaction [71] and more depressive symptoms [72]. Likewise, the positive effect of both parents’ age on their job satisfaction aligns with previous research reporting that age is positively associated with job satisfaction [73]. Our findings that mothers who work for themselves are more satisfied with their jobs than mothers who are employed are consistent with earlier research that shows self-employed people are more satisfied with their jobs because they have more freedom, flexibility, individual responsibility for finishing tasks, safe working environments, and a friendly atmosphere [74].

The limitations of this study should be addressed in future research. First, this study design was cross-sectional; the so-called effects derived from the Actor-Partner Interdependence Model refer to effects; the cross-sectional design of this study cannot establish causality. Second, the findings in this study cannot be generalized to Chilean dual-earner parents with adolescents due to some of the sample characteristics. Specifically, the sample was non-probabilistic and self-selected, representing a particular family structure with a relatively larger size than data from the Chilean 2017 National Census (an average of four family members, including two children). Third, since this study design predates the pandemic, specific conditions related to lockdown measures were not assessed, namely whether parents were working remotely; hence, the relationships found here cannot be related to working conditions. Fourth, this study did not evaluate the direct involvement of adolescents in their parents’ family-to-work conflict, which stems from behavioral difficulties in the workers’ children [8], and adolescents present novel challenges to their parents related to their conflicting needs for support and autonomy. Future studies will benefit from exploring children’s role in their parents’ negotiations at the work-family interface. This is important because, in the post-Covid era, parents need help addressing their children’s social and emotional challenges [75]. Furthermore, more information is required to determine other variables that contribute to the connections found in this study, i.e., the workplace conditions (e.g., those derived from formal employment vs. self-employment) and family dynamics that may be involved in strengthening or weakening these links for dual-earner parents. 

## 5. Conclusions

Our findings contribute to the work-family interface literature by showing the potential effects of family-to-work conflict (FtoWC) on psychological distress and job satisfaction in different-sex dual-earner parents. In this sample, both parents had similar levels of job satisfaction, whereas mothers reported higher family-work conflict and psychological distress than fathers. A higher FtoWC was linked to higher psychological distress and lower job satisfaction, regardless of the parent’s gender. In contrast, only the fathers’ psychological distress directly reduced their job satisfaction. Although no direct crossover effects from one parent’s FtoWC to the other’s job satisfaction were found, our main contribution to knowledge is that fathers and mothers experienced a double negative indirect effect on their job satisfaction from their own and their partner’s FtoWC via the fathers’ psychological distress. Our results also suggest the importance of identifying variables that could mediate between family-to-work conflict and job satisfaction because the direct effect of family-to-work conflict on job satisfaction may be low in the presence of other mediators. Furthermore, our findings suggest gender differences in how parents managed the boundaries between work and family during the pandemic. 

The theoretical contribution of these results is found in the combined use of the COR and the boundary theories. Results in fathers were consistent with the loss spiral of resources posited by the COR. However, the COR theory alone failed to explain the lack of a spiral of resource loss in mothers because their job satisfaction was not affected by their psychological distress. On the other hand, the boundary theory allowed for the explanation of the differences between mothers and fathers based on the different ways of interaction between the family and work domains that may be adopted by parents. Taken together, these findings support the principles of these theories, but they also indicate the need to explore a larger theoretical framework to account for gender in the work-family interface. Although the COVID-19 pandemic was declared over, our findings are interesting for future health crises. Our study shows that job satisfaction and psychological distress scores in mothers and fathers were similar to those obtained in studies in Chile before the pandemic [17]. This underscores the need for further research to explore family-to-work conflict and its related outcomes after the pandemic and possible coping strategies that parents use so that no significant changes are discerned in their psychological distress and job satisfaction before and after the pandemic. We also propose further studying the conditions of different-sex dual-earner parents, considering that family-to-work conflict outcomes may be dyadic. As these work-home dynamics entail the generation, transmission, and loss of resources, future studies may also focus on identifying family demands and resources that heighten and alleviate both parents’ FtoWC and their impact on satisfaction in other life domains, such as family. Furthermore, more research is needed to better understand the differences in managing the boundaries between work and family according to the parent’s gender. In our study, the family’s socioeconomic status, both parents’ ages, and the mother’s type of employment as control variables significantly affected psychological distress and job satisfaction. Future studies should also assess the moderating role of these variables. These proposed studies should also include families of diverse sizes and compositions and consider how these families relate to gender-related roles and values (i.e., traditional vs. egalitarian gender approaches) to better understand family-to-work phenomena and experiences on a larger scale.

These results also have practical implications. Our results on psychological distress indicate the need for employers to consider policies and training that help workers manage their multiple life roles. Although women experience higher psychological distress than men, policymakers and organizations should provide both parents with resources to reduce their psychological distress, particularly during health crises. Policymakers and organizations should give women resources to reduce the pressure, not just from domestic tasks but also from homeschooling during health crises, since mothers faced increased family-to-work conflict that directly affected both fathers’ and mothers’ psychological distress. Addressing workers’ subjective well-being and domain satisfaction requires providing resources at the personal (e.g., stress management, flexible schedules) and organizational (e.g., family-friendly policies, support from supervisors and co-workers) levels. To further comprehend the gender-based constraints that men and women face in the work and family spheres, these interventions also require a gender perspective. Interventions and training should emphasize men’s varied responsibilities to promote and encourage men’s participation in their home and family lives and help reduce women’s work-family conflict. In addition, training in managing the boundaries between work and family may be relevant, particularly for men. Lastly, interventions should be considered for the family as a whole, including children, given that family members are interconnected via mutual interactions. Therefore, although children were not included in the present study, adolescents are a source of emotional and instrumental support for their working parents, which may potentially reduce their parents’ family-to-work conflict and psychological distress. 

These recommendations were not only relevant during the pandemic. The need to manage work-family conflict is still important for dual-earner parents’ well-being [76], and the symptomatology of psychological distress is still relatively high in workers who returned to the workplace [77] in the post-pandemic context. Furthermore, although evidence from China [78] suggests that job satisfaction will rise post-pandemic for those workers who commute to their workplace, other authors have shown that workload reduced employees’ job satisfaction during the period of return to working on-site [79]. Therefore, organizations should also manage their employees’ workload during this new stage.

## Figures and Tables

**Figure 1 behavsci-14-00056-f001:**
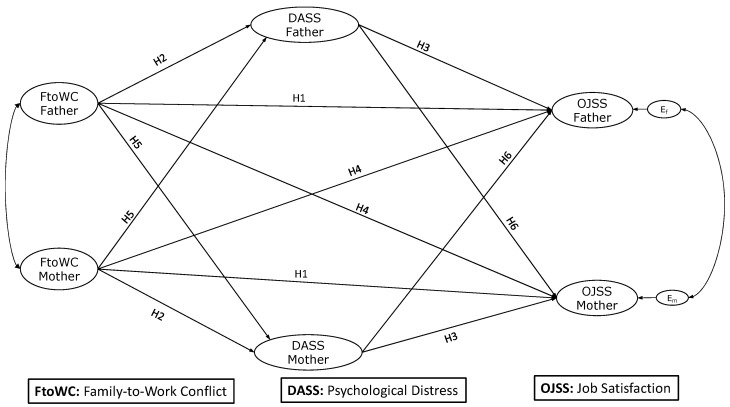
Conceptual model of the proposed actor and partner effects in dual-earner parents with adolescent children. Note: E_m_ and E_f_: residual errors on OJSS for the mothers and fathers, respectively. The indirect effects of Psychological Distress (H7) were not shown in the conceptual path diagram to avoid cluttering the figure.

**Figure 2 behavsci-14-00056-f002:**
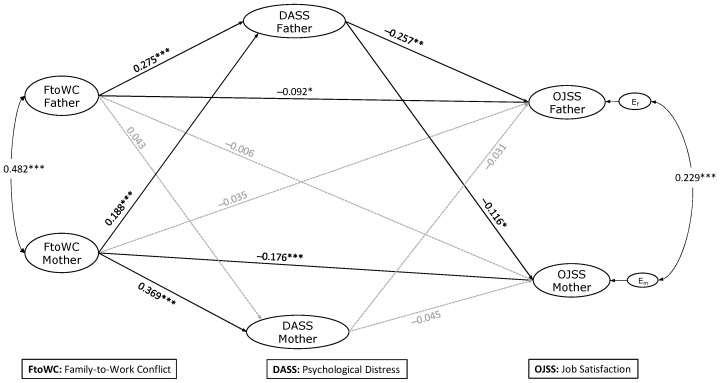
Actor-partner interdependence model in dual-earner parents with adolescent children. Note: E_m_ and E_f_: residual errors on OJSS for mothers and fathers, respectively. The control for the effects of both parents’ age, type of employment, and number of working hours, as well as the family SES and the number of children, on the dependent variables of both parents (DASS and OJSS), were not shown in the path diagram. * *p* < 0.05; ** *p* < 0.01; *** *p* < 0.001.

**Table 1 behavsci-14-00056-t001:** Sample characteristics of participant dual-earner couples.

Characteristic	Temuco (*n* = 430)	Rancagua (*n* = 430)	Total Sample	*p*-Value
Age [Mean (SD)] ^a^				
Father	42.0 (8.6)	42.3 (7.8)	42.2 (8.2)	0.607
Mother	38.6 (7.3)	39.4 (6.6)	39.0 (6.9)	0.091
Number of family members [Mean (SD)] ^a^	4.3 (1.1)	4.3 (1.0)	4.3 (1.0)	0.921
Number of children [Mean (SD)] ^a^	2.2 (0.9)	2.2 (0.9)	2.2 (0.9)	0.525
Socioeconomic status (SES) (%) ^b^				
High	1.6	3.7	2.7	<0.001
Middle	74.9	83.0	79.0	
Low	23.5	13.3	18.4	
Mothers’ type of employment (%) ^b^				
Employee	68.1	62.8	65.5	0.099
Self-employed	31.9	37.2	34.5	
Fathers’ type of employment (%) ^b^				
Employee	73.5	75.3	74.4	0.532
Self-employed	26.5	24.7	25.6	
Mothers’ working hours (%) ^b^				
45 h per week	48.4	44.0	46.2	0.194
Less than 45 h per week	51.6	56.0	53.8	
Fathers’ working hours (%) ^b^				
45 h per week	69.8	67.2	68.5	0.419
Less than 45 h per week	30.2	32.8	30.2	

*N* = 860. ^a^ Independent sample *t*-test. ^b^ *p*-value corresponds to the (bilateral) asymptotic significance obtained in Pearson’s Chi-square Test.

**Table 2 behavsci-14-00056-t002:** Descriptive statistics and correlations for Family-to-Work (FtoWC), Psychological Distress (DASS), and Job Satisfaction (OJSS) in different-sex dual-earner parents with adolescent children.

	M (SD)	Correlations					
		1	2	3	4	5	6
1. Mothers’ FtoWC	7.22 (3.16)	1	0.395 **	0.344 **	0.250 **	−0.193 **	−0.127 **
2. Fathers’ FtoWC	6.44 (2.93)		1	0.185 **	0.305 **	−0.117 **	−0.149 **
3. Mothers’ DASS	30.47 (10.43)			1	0.376 **	−0.145 **	−0.139 **
4. Fathers’ DASS	27.21 (8.44)				1	−0.144 **	−0.226 **
5. Mothers’ OJSS	22.30 (5.06)					1	0.274 **
6. Fathers’ OJSS	22.10 (4.68)						1

Note: *N* = 860. ** *p* < 0.01; two-tailed.

**Table 3 behavsci-14-00056-t003:** Bias-corrected confidence intervals of specific mediation effects of a parent’s psychological distress (DASS).

Specific Indirect Effects	Estimate	Lower 2.5%	Upper 2.5%	*p*-Value
Fathers’ FtoWC → Fathers’ DASS → Fathers’ OJSS	−0.071	−0.091	−0.031	<0.001 ***
Mothers’ FtoWC → Mothers’ DASS → Mothers’ OJSS	−0.017	−0.047	0.014	0.298
Mothers’ FtoWC → Fathers’ DASS → Mothers’ OJSS	−0.032	−0.041	−0.002	0.032 *
Fathers’ FtoWC → Mothers’ DASS → Fathers’ OJSS	−0.001	−0.005	0.003	0.574
Mothers’ FtoWC → Fathers’ DASS → Fathers’ OJSS	−0.048	−0.075	−0.018	0.001 **
Mothers’ FtoWC → Mothers’ DASS → Fathers’ OJSS	−0.011	−0.041	0.019	0.477
Fathers’ FtoWC → Fathers’ DASS → Mothers’ OJSS	−0.028	−0.054	−0.002	0.037 *
Fathers’ FtoWC → Mothers’ DASS → Mothers’ OJSS	−0.002	−0.006	0.003	0.442

Note: FtoWC: Family-to-work conflict; OJSS: Job satisfaction. * *p* < 0.05; ** *p* < 0.01; *** *p* < 0.001.

## Data Availability

The original contributions presented in the study are included in the article, further inquiries can be directed to the corresponding author.

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
