# Peer review of "Intra- and Inter-Individual Associations of Family-to-Work Conflict, Psychological Distress, and Job Satisfaction: Gender Differences in Dual-Earner Parents during the COVID-19 Pandemic"

_behavsci, 2024, doi:10.3390/bs14010056_

Round 1
Reviewer 1 Report
Comments and Suggestions for Authors
This is an interesting study. Authors need to compare the result during the COVID and after the COVID in order to justify the argument. Please give me a background of conservation of resources theory. Source of family work conflicts come from kids. What are their views? Please explain actor partner interdependence model in detail. Pleas incorporate moderating factors in your model: working hours, type of employment, number of family members and economic status.
Comments on the Quality of English LanguageVery difficult to read the paper
Author Response
We appreciate all your remarks and suggestions in order to improve the manuscript. We look forward to hearing from you and we are willing to make further changes should it be considered necessary. All the changes are in red in the new version of the manuscript.
This is an interesting study. Authors need to compare the result during the COVID and after the COVID in order to justify the argument.
R: Thank you four valuable suggestion. However, as the objectives of the study did not include to compare results during and after COVID, we discussed that the practical implication of the study are also valid for the post-COVID stage (Line 877-885, 814-815).
Please give me a background of conservation of resources theory.
R: Thank you for your suggestion. Accordingly, background of the COR theory has been added (Line 137-145).
Source of family work conflicts come from kids. What are their views?
R: Thank you for your question. This is one of the limitations of the study, therefore we suggested that future studies will benefit from exploring children’s role in their parents’ negotiations in the work-family interface (Line 809-814).
Please explain actor partner interdependence model in detail.
R: Thank you for your suggestion. Accordingly, more detail about the APIM has been added (Line 133-1358, 441-449).
Pleas incorporate moderating factors in your model: working hours, type of employment, number of family members and economic status.
R: Thank you very much for your suggestion. However, we used these variables as control variables and only socioeconomic status affected mothers’ DASS and job satisfaction, while only the mothers’ type of employment significantly affected their own job satisfaction (line 531-537, 788-798). However, we added a phrase suggesting that future studies should also assess the moderating role of these variables (Line 849-852).
Comments on the Quality of English Language: Very difficult to read the paper
R: Thank you for your suggestion. A native English speaker re-reviewed the entire manuscript.
Reviewer 2 Report
Comments and Suggestions for Authors
Comments to the authors
It is an original manuscript with an interesting research report with very good organization and readability. However, it has a few issues in its construction that need to be solved, as noted below. I encourage authors to engage in improving the manuscript, which presents a potential contribution.
The Introduction section has issues in its construction that need to be solved, as noted below.
a) The conceptual framework needs to be expanded. Add the discussion of the following concepts in the Introduction section: gender, gender equality, pandemic issues, and its impact. Moreover, try further to contextualize the conceptual framework in the Chilean context, pointing out similarities and incongruities between cultures to indicate what would be universal and what would be local in this conceptualization.
b) The authors should provide a more detailed description of Chilean sociocultural backgrounds, especially regarding gender stereotypes, education and the world of work features, and other cultural aspects. That is necessary to contextualize the discussion of the data. The study may reach mistaken conclusions if this sociocultural, gender, work, and educational contextualization is not provided.
c) Present further secondary data from the Chilean context to underpin the conceptual claims presented.
d) Include a summary of the main ideas at the end of the Introduction.
e) It would be important to summarize the studies on the same subject at two points in time: before the pandemic and after the pandemic to check the impact of the pandemic on the psychosocial phenomenon studied.
In the Method, the authors provide suitable information regarding the required statistical analysis. The number of additional variables collected was satisfactory, and the analysis was appropriate. However, it has a few issues in its construction that need to be solved, as noted below.
a) Briefly describe how the sample specificities (variables) were controlled and discussed in the results.
b) Lack of a more detailed characterization of participants and a brief description of the context studied.
c) Methodological issue: it is necessary to present data indicating the epistemological coherence of the conceptual bases of the instruments used in the research.
The authors presented the report of the results reasonably.
My third concern relates to the discussion section.
a) The discussion needs to contrast the pre- and post-pandemic periods and highlight what is specific to the Chilean context and what can be generalized in terms of the subject studied to foster international interest in the manuscript.
b) The study seems to be more descriptive than explanatory. Include some discussions and analytical hypotheses to make the manuscript more explanatory, improving its quality and relevance.
c) The practical implications could be extended.
Author Response
We appreciate all your remarks and suggestions in order to improve the manuscript. We look forward to hearing from you and we are willing to make further changes should it be considered necessary. All the changes are in red in the new version of the manuscript.
Comments to the authors
It is an original manuscript with an interesting research report with very good organization and readability. However, it has a few issues in its construction that need to be solved, as noted below. I encourage authors to engage in improving the manuscript, which presents a potential contribution.
R: Thank you very much for your comments and suggestions.
The Introduction section has issues in its construction that need to be solved, as noted below.
a) The conceptual framework needs to be expanded. Add the discussion of the following concepts in the Introduction section: gender, gender equality, pandemic issues, and its impact. Moreover, try further to contextualize the conceptual framework in the Chilean context, pointing out similarities and incongruities between cultures to indicate what would be universal and what would be local in this conceptualization.
R: Thank you very much for your suggestions. Gender, gender equality, pandemic issues, and its impacts have been added to the introduction (Line 60-62, 69-71, 81-89, 107-121, 323-346). For each intraindividual hypothesis we anticipated if we expect a more local or universal result (line 187-190, 202-207,215-217), while for the interindividual hypotheses we anticipated more unidirectional crossover effects from men to women (Line 271-273).
b) The authors should provide a more detailed description of Chilean sociocultural backgrounds, especially regarding gender stereotypes, education and the world of work features, and other cultural aspects. That is necessary to contextualize the discussion of the data. The study may reach mistaken conclusions if this sociocultural, gender, work, and educational contextualization is not provided.
R: Thank you for your suggestion. The requested information has been added (Line 107-121, 323-346).
c) Present further secondary data from the Chilean context to underpin the conceptual claims presented.
R: Thank you for your suggestion. Secondary data from Chile has been added (Line 107-121, 323-346).
d) Include a summary of the main ideas at the end of the Introduction.
R: Thank you for your suggestion. Accordingly, a paragraph that summarise the hypotheses has been added at the end of the introduction (Line 350-362).
e) It would be important to summarize the studies on the same subject at two points in time: before the pandemic and after the pandemic to check the impact of the pandemic on the psychosocial phenomenon studied.
R: Thank you four valuable suggestion. However, as the objectives of the study did not include to compare results during and after COVID, we discussed that the practical implication of the study are also valid for the post-COVID stage (Line 788-798, 877-885).
In the Method, the authors provide suitable information regarding the required statistical analysis. The number of additional variables collected was satisfactory, and the analysis was appropriate. However, it has a few issues in its construction that need to be solved, as noted below.
a) Briefly describe how the sample specificities (variables) were controlled and discussed in the results.
R: Thank you for your suggestions. The phrase in which the use of control variables has been reworded to better explain the procedure (Line 464-467). The discussion of the results regarding the control variables that significantly affected the model have been added (Line 788-798).
b) Lack of a more detailed characterization of participants and a brief description of the context studied.
R: Thank you for your suggestion. A more detailed characterization of the participants has been added based on data presented in Table 1 (Line 483-487) and a brief description of the regions in where the cities are located (where the data was collected) has been incorporated (Line 367-374).
c) Methodological issue: it is necessary to present data indicating the epistemological coherence of the conceptual bases of the instruments used in the research.
R: Thank you for your suggestion. This data has been added for each scale (Line 397-399, 417-419, 428-430).
The authors presented the report of the results reasonably.
R: Thank you for your comment. In response to your suggestion regarding the hypotheses and a request of the editor, changes have been made in the wording of the hypotheses (Line 191-192, 208-209, 218-219, 239-240, 254-255, 269-270, 312-313) results (Line 508-530, 552) and discussion (Line 565-798).
My third concern relates to the discussion section.
a) The discussion needs to contrast the pre- and post-pandemic periods and highlight what is specific to the Chilean context and what can be generalized in terms of the subject studied to foster international interest in the manuscript.
R: Thank you four valuable suggestion. However, as the objectives of the study did not include to compare results during and after COVID, we include discussed that the practical implication of the study are also valid for the post-COVID stage (Line 788-798, 877-885).
b) The study seems to be more descriptive than explanatory. Include some discussions and analytical hypotheses to make the manuscript more explanatory, improving its quality and relevance.
R: Thank you for your suggestion. In response to your suggestion regarding the hypotheses and a request of the editor, changes have been made in the wording of the hypotheses (Line 191-192, 208-209, 218-219, 239-240, 254-255, 269-270, 312-313) results (Line 508-530, 552) and discussion (Line 565-798), to make the manuscript more explanatory.
c) The practical implications could be extended.
R: Thank you for your suggestion. Accordingly, they have been expanded (Line 856-885).
Reviewer 3 Report
Comments and Suggestions for Authors
The paper provides insightful evidence about intra- and interindividual associations of family-to-work conflict, psychological distress, and job satisfaction, considering the gender differences in dual-earner parents during the covid-19 pandemic. Anyway, there are some issues that still need some improvement. I encourage the authors to consider the comments given below and revise the paper accordingly in order to enhance the overall quality and completeness of the paper.
(1) At the beginning of the introduction, it will be good if the authors could provide the some more overview about the covid-19 crisis. In particular, some more recent paper about the psychological impacts of covid-19 pandemic should be included as the additional references. I recommend the following papers to be included when reviewing the psychological impacts of covid-19 pandemic.
· How Does Mindfulness Help University Employees Cope with Emotional Exhaustion during the COVID-19 Crisis? The Mediating Role of Psychological Hardiness and the Moderating Effect of Workload, Scandinavian Journal of Psychology. 63(5), 449-461. https://doi.org/10.1111/sjop.12826
· "Pedagogical transitions experienced by higher education faculty members – “Pre-Covid to Covid”", Journal of Applied Research in Higher Education, Vol. 14 No. 3, pp. 987-1006. https://doi.org/10.1108/JARHE-01-2021-0028
(2) In page 3, the authors indicate that evidence of the relationship between FtoWC and job satisfaction is mixed. It will be helpful to provide some reason why some study did not find significant correlation between FtoWC and job satisfaction
(3) In the paragraph that explains the COR theory, the authors should provide some examples of research that apply the COR theory to the covid-19 context. I recommend the following papers to be included as the references when reviewing the COR theory.
· Effects of Trust in Organizations and Trait Mindfulness on Optimism and Perceived Stress of Flight Attendants during the COVID-19 Pandemic, Personnel Review, 52(3), 882-899. https://doi.org/10.1108/PR-06-2021-0396
· What makes one feel eustress or distress in quarantine? An analysis from conservation of resources (COR) theory. Br J Health Psychol, 26: 606-623. https://doi.org/10.1111/bjhp.12501
· Careers after COVID-19: challenges and changes, Human Resource Development International, 23:4, 427-437, https://doi.org/10.1080/13678868.2020.1779576
Author Response
We appreciate all your remarks and suggestions in order to improve the manuscript. We look forward to hearing from you and we are willing to make further changes should it be considered necessary. All the changes are in red in the new version of the manuscript.
The paper provides insightful evidence about intra- and interindividual associations of family-to-work conflict, psychological distress, and job satisfaction, considering the gender differences in dual-earner parents during the covid-19 pandemic. Anyway, there are some issues that still need some improvement. I encourage the authors to consider the comments given below and revise the paper accordingly in order to enhance the overall quality and completeness of the paper.
R: Thank you very much for your comments and suggestions.
(1) At the beginning of the introduction, it will be good if the authors could provide the some more overview about the covid-19 crisis. In particular, some more recent paper about the psychological impacts of covid-19 pandemic should be included as the additional references. I recommend the following papers to be included when reviewing the psychological impacts of covid-19 pandemic.
R: Thank you for your suggestion. Both studies suggested have been incorporated (Line 52-53, 63-65).
- How Does Mindfulness Help University Employees Cope with Emotional Exhaustion during the COVID-19 Crisis? The Mediating Role of Psychological Hardiness and the Moderating Effect of Workload, Scandinavian Journal of Psychology. 63(5), 449-461. https://doi.org/10.1111/sjop.12826
- "Pedagogical transitions experienced by higher education faculty members – “Pre-Covid to Covid”", Journal of Applied Research in Higher Education, Vol. 14 No. 3, pp. 987-1006. https://doi.org/10.1108/JARHE-01-2021-0028
(2) In page 3, the authors indicate that evidence of the relationship between FtoWC and job satisfaction is mixed. It will be helpful to provide some reason why some study did not find significant correlation between FtoWC and job satisfaction
R: Thank you for your suggestion. Explanation for the null association between FtoWC and job satisfaction has been added for two studies (Line x-x).
(3) In the paragraph that explains the COR theory, the authors should provide some examples of research that apply the COR theory to the covid-19 context. I recommend the following papers to be included as the references when reviewing the COR theory.
R: Thank you for your suggestion. The background of the COR theory has been extended according to the Reviewer’s 1 suggestion and the three studies suggested have been incorporated (Line 145-155).
- Effects of Trust in Organizations and Trait Mindfulness on Optimism and Perceived Stress of Flight Attendants during the COVID-19 Pandemic, Personnel Review, 52(3), 882-899. https://doi.org/10.1108/PR-06-2021-0396
- What makes one feel eustress or distress in quarantine? An analysis from conservation of resources (COR) theory. Br J Health Psychol, 26: 606-623. https://doi.org/10.1111/bjhp.12501
- Careers after COVID-19: challenges and changes, Human Resource Development International, 23:4, 427-437, https://doi.org/10.1080/13678868.2020.1779576
Reviewer 4 Report
Comments and Suggestions for Authors
My only suggestion is to correct the mall typo of al-lowed, which can be cleaned in copyediting as well. The manuscript addresses the concerns presented by the reviewer(s).
Round 2
Reviewer 1 Report
Comments and Suggestions for Authors
Thank you for your responses. Some of my concerns were not resolved in the revised paper including the comparative study, moderating factor and kids' perspective.
My main point is the authors did not resolve my concerns. For example, to compare the result of COVID-19 and after COVID-19; and the perspective from the kids. It shall take some time authors to check their data collection. I don't think changing or revising the paper based on existing data could reach our standard.
In their revised version, efforts has been made to revise the introduction and hypotheses testing. How about hypothesis 3 explanation?
Also there is a mess in the reference list, No. 60, 61, 67 and 68 do not appear in the text.
How about the theoretical contribution?
Comments on the Quality of English LanguageEnglish was improved.
Author Response
We again appreciate your remarks and suggestions in order to improve the manuscript. The changes are in red in the new version of the manuscript (unmarked).
Thank you for your responses. Some of my concerns were not resolved in the revised paper including the comparative study, moderating factor and kids' perspective.
R: Thank you for your comments and suggestion to improve the manuscript. Regarding the moderating variables, the aim of this study was to examine the relations between FtoWC, DASS and Job Satisfaction, and thus the variables suggested as moderators were controlled for in the original analysis to observe clearer pathways in the model. For this reason, we suggest that these variables are included as moderators in future studies, as stated in the first revision.
My main point is the authors did not resolve my concerns. For example, to compare the result of COVID-19 and after COVID-19; and the perspective from the kids. It shall take some time authors to check their data collection. I don't think changing or revising the paper based on existing data could reach our standard.
R: We apologize in regards to the comparison of the results during and after COVID-19. This comparison was not part of our objectives, but we again conducted a search in WOS to find published papers to comply with this requirement. However, our search did not return any manuscripts published post-Covid with the variables used in the model, with which we could compare our results. This may be because the pandemic was declared over in May of this year and any potential manuscripts on the topic might still be under development. However, if the reviewer suggested some papers, we will of course cite them to improve our discussion. Regarding adolescents, the aim of our paper was not to explore their perspective regarding their parent’s FtoWC, therefore we don’t have a measure in this regard. We acknowledged this issue as one of the limitations of the study in the original version of the manuscript, and therefore we suggested that future studies will benefit from exploring children’s role in their parents’ negotiations in the work-family interface.
In their revised version, efforts has been made to revise the introduction and hypotheses testing. How about hypothesis 3 explanation?
R: Thank you for your question, but the explanation of the H3 is in lines 621-655, and it has also been modified in comparison to the original version of the manuscript.
Also there is a mess in the reference list, No. 60, 61, 67 and 68 do not appear in the text.
R: Thank you very much for your rigor when revised the references. However, references 60, 61 and 67 were cited in the text (lines 329, 331, 340 , 442). Thank you for noticing our mistake regarding reference No. 68, the first reference 69 in the text belongs to reference 68 (line 472).
How about the theoretical contribution?
R: Thank you for your question. A theoretical contribution has been added (Line 834-842).
Reviewer 3 Report
Comments and Suggestions for Authors
The authors have successfully addressed all comments. The quality of the paper is now adequate for the publication.
Author Response
We again appreciate your remarks and suggestions in order to improve the manuscript
Round 3
Reviewer 1 Report
Comments and Suggestions for Authors
Some improvements made. Thanks for your responses and treat some of my suggestions out of the scope of study, as limitations of the study and further research areas.